# COVID-19 Infection Fatality Rate Associated with Incidence—A Population-Level Analysis of 19 Spanish Autonomous Communities

**DOI:** 10.3390/biology9060128

**Published:** 2020-06-16

**Authors:** Chris Kenyon

**Affiliations:** 1HIV/STI Unit, Institute of Tropical Medicine, 2000 Antwerp, Belgium; ckenyon@itg.be; Tel.: +32-3-2480796; Fax: +32-3-2480831; 2Division of Infectious Diseases and HIV Medicine, University of Cape Town, Cape Town 7700, South Africa

**Keywords:** SARS CoV-2, COVID-19, case fatality rate, epidemiology, infection fatality rate

## Abstract

Previous studies have found large variations in the COVID-19 infection fatality rate (IFR). This study hypothesized that IFR would be influenced by COVID-19 epidemic intensity. We tested the association between epidemic intensity and IFR using serological results from a recent large SARS-CoV-2 serosurvey (N = 60,983) in 19 Spanish regions. The infection fatality rate for Spain as a whole was 1.15% and varied between 0.13% and 3.25% in the regions (median 1.07%, IQR 0.69–1.32%). The IFR by region was positively associated with SARS-CoV-2 seroprevalence (rho = 0.54; *p* = 0.0162), cases/100,000 (rho = 0.75; *p* = 0.002), hospitalizations/100,000 (rho = 0.78; *p* = 0.0001), mortality/100,000 (rho = 0.77; *p* = 0.0001) and case fatality rate (rho = 0.49; *p* = 0.0327). These results suggest that the SARS-CoV-2 IFR is not fixed. The Spanish regions with more rapid and extensive spread of SARS-CoV-2 had higher IFRs. These findings are compatible with the theory that slowing the spread of COVID-19 down reduces the IFR and case fatality rate via preventing hospitals from being overrun, and thus allowing better and lifesaving care.

Roques et al., recently estimated in this journal that the adjusted SARS-CoV-2 infection fatality rate (IFR) in France was 0.8% [1]. Other studies have found large variations in estimated IFR. A systematic review and metanalysis found an aggregated IFR of 0.75% (95% CI 0.49–1.01%) with an extremely high heterogeneity (I^2^ exceeding 99%; *p* < 0.0001) [2]. One of the best quality estimates of IFR was determined in a relatively heavily affected community in Germany by accurately determining cause of death and establishing exposure via combined serological and nucleic acid amplification testing of a representative sample of the population [3]. This study found the crude IFR to be 0.37% and the adjusted IFR 0.28% (N = 909). Estimates in the United States have varied from 0.17% in Santa Clara County California [4] to 1.3% in the whole United States [5]. These studies used different sampling and serological tests and were conducted at different stages of the local epidemics, making comparisons between studies problematic. Furthermore, evaluations of the related COVID-19 case fatality rates (CFR), have suggested that CFR was dependent on the efficacy of local response efforts [6,7]. A study from China, for example, found that the timely supply of adequate medical resources lowered the CFR from around 4.5% to 0.5% [8]. The authors suggested that high CFRs were in part determined by hospital capacity being exceeded and, as a result, patients receiving suboptimal care [6,7]. A ‘flattening the curve’ approach has been advocated to prevent this from happening [6,9,10]. In this paper I hypothesized that IFR would similarly be influenced by four dimensions of epidemic intensity–seroprevalence of SARS-CoV-2, incidence of COVID-19 cases, COVID-19 mortality and COVID-19 hospitalizations.

I tested the association between IFR and these variables using serological results from a recent large SARS-CoV-2 serosurvey (N = 60,983) in Spain’s 17 autonomous communities (Comunidades Autonomas), and two autonomous cities of Spain (henceforth termed Spain’s 19 regions) [11]. The survey was conducted between 27 April 2020 and 11 May 2020. 74.7% of those contacted agreed to participate in the survey. The data for COVID-19-related clinical cases, deaths and hospital admissions/100,000 per region were obtained from other Spanish Ministry of Health documents reporting data for the epidemic situation as of 13 May 2020 [12]. See online supplement for complete list of sources of information as well as the definitions of variables used. All the data sources are open-access and only aggregated ecological level data were used. As such, no ethical clearance was required. Of note, the Spanish epidemic was declining in all regions from late March, and by 4 April 2020 the effective reproductive number was below one in all regions (https://cnecovid.isciii.es/covid19/#declaraci%C3%B3n-agregada).

The IFR per region was calculated as the percent of individuals who died from COVID-19 as of 13 May 2020 divided by the percent of individuals seropositive for COVID-19 in the seroprevalence survey (conducted 27 April 2020 and 11 May 2020). Spearman’s correlation was used to assess the association between regional IFR and CFR, seroprevalence of SARS-CoV-2, incidence of COVID-19 cases/100,000, COVID-19 mortality/100,000 and COVID-19 hospitalizations/100,000. A sensitivity analysis was conducted in which these correlations were repeated, controlling for the percent of the population aged 65 years or older. Linear regression was used for these analyses.

Complete data was available for all 19 regions. The infection fatality rate for Spain as a whole was 1.15%, and varied between 0.13% and 3.25% in the regions (median 1.07%, IQR 0.69–1.32%; Figure 1; Appendix A). There was also considerable variation in the crude CFR (median 10.5% IQR 8.7–13.0%), the number of COVID-19 cases/100,000 (median 387, IQR 184–953 cases/100,000), COVID-19 mortality/100,000 (34.5, IQR 15.8–75.4) and the number of COVID-19 hospitalizations/100,000 (median 173, IQR 72–335/100,000).

The IFR was positively associated with seroprevalence (rho = 0.54; *p* = 0.0162), cases/100,000 (rho = 0.75; *p* = 0.002), hospitalizations/100,000 (rho = 0.78; *p* = 0.0001), mortality/100,000 (rho = 0.77; *p* = 0.0001) and CFR (rho = 0.49; *p* = 0.0327; Figure 1). The IFR was also positively associated with the percent aged 65 and older (rho = 0.68; *p* = 0.0014).

In the sensitivity analyses controlling for the percent of the population aged 65 or older, the IFR remained significantly associated with cases/100,000, hospitalizations/100,000 and mortality/100,000 (all *p* ≤ 0.05; Appendix A). In these multivariate analyses, the percent aged 65 or older was significantly positively associated with all five variables assessed (all *p* ≤ 0.05; Appendix A).

These results suggest that the SARS-CoV-2 IFR is not fixed. The Spanish regions with more rapid and extensive spread of SARS-CoV-2 had higher IFRs. These findings are compatible with the theory that slowing the spread of COVID-19 down reduces the IFR and CFR via preventing hospitals from being overrun and thus allowing better and lifesaving care [8]. These findings thus support the findings from other studies that flatten-the-curve strategies such as physical distancing, test-treat-isolate, and cocooning high-risk populations reduce not only the total number of cases and deaths but also the probability of deaths per infection [7,13]. They thus provide further evidence that control strategies that aim for achieving herd immunity in the short-term are likely to result in excess mortality.

The percent of people of age 65 years or older was a positive predictor of IFR and independently associated with seroprevalence in the multivariate analyses. Although there are a number of explanations for these findings, they are parsimoniously explained by a high proportion of elderly being a predictor of incidence and mortality amongst those infected. The higher COVID-19 CFR/IFR amongst the elderly is well established [3,13,14]. Our results are consistent with the evidence that the incidence of COVID-19 is higher in elderly populations [15]. Since severe COVID-19 is characterized by high viral load in respiratory secretions, and this translates into increased infectiousness, vulnerable older populations confined to close spaces, such as elderly-care-type facilities, may have higher COVID-19 incidence [16].

A strength of this analysis is that it is based on a large representative sample of the whole of Spain that experienced a heterogenous epidemic but used the same methodology to assess seroprevalence and cause of death throughout the country. As noted above, this was not possible with previous IFR studies which used different methodologies.

There are however a number of limitations to this analysis. This study only controlled for age structure [14]. It is possible that the regions with higher IFRs had a higher prevalence of comorbidities associated with severe COVID-19 [14]. The estimates of the IFR estimates could be biased downwards by underreporting COVID-19 deaths or a failure to account for the delay from diagnosis to death [13,14]. The fact that we conducted our analyses using deaths from late in the epidemic should however help to minimize this bias. The number of COVID-19 deaths in two of the regions was small, which makes the mortality rate estimates from these regions susceptible to large fluctuations with small changes in the number who died (Appendix A). Removing these two regions did not, however, substantively change the results (Appendix A). A strong correlation between the IFR and CFR was found. The CFR could be considered to represent the upper bound of the IFR.

Despite these methodological weaknesses, this study provides further support for the concept that the IFR for SARS-CoV-2 is dependent on epidemic intensity and that interventions that slow the spread of SARS CoV-2 will reduce infection fatality rates.

## Figures and Tables

**Figure 1 biology-09-00128-f001:**
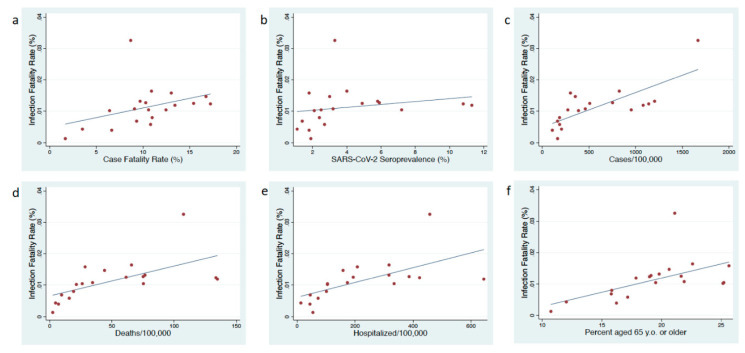
Scatter plot of infection fatality rate versus case fatality rate (**a**), seroprevalence of SARS-CoV-2 (**b**), cumulative number of cases of COVID-19/100,000 (**c**), COVID-19 attributable mortality/100,000 (**d**) and COVID-19 attributable hospitalization/100,000 (**e**) in 19 Spanish regions (**f**) percent of the population 65 years or older.

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
