# Peer review of "COVID-19 Infection Fatality Rate Associated with Incidence—A Population-Level Analysis of 19 Spanish Autonomous Communities"

_biology, 2020, doi:10.3390/biology9060128_

Round 1

Reviewer 1 Report

In this letter, the author shows that the infection fatality rate (IFR) depends on the epidemic intensity, by analysing the dependence of the ratio between the number of deaths and the seroprevalence (= the IFR) with respect to the seroprevalence and other measures of the epidemic intensity in 19 regions in Spain. 

Other studies had already reported a positive association between the case fatality rate (CFR) and the epidemic intensity. However, the interpretation of this finding is difficult, as it could be due to a saturation of the medical care structures, as emphasized by the author, but it could also be simply a consequence of the definition of the CFR (number of deaths/number of reported cases), and of the testing strategy. For instance, if the number of tests is limited, the CFR automatically increases with the prevalence of the disease. 

The IFR is much more informative, and should not depend on the testing strategy. Thus, this study shows convincingly that the fatality rate does depend on the epidemic intensity. 

Overall, the letter is well-written, and contains simple but important findings. I therefore recommend its publication. Please consider my small edits below. 

  1. The effective reproduction number cannot be negative (l. 41). Probably the author means that it is below 1.
  2. It could be interesting to discuss the links between the CFR and the IFR. In particular, a lower bound for the IFR is the number of deaths divided by the total population in the region, and an upper bound is the CFR. 
  3. The author indicates that he did not control for any confounders. It could be nice to check for a correlation between the median age (or the proportion of people older than some age) and the IFR in the considered regions. Such data can be found here : https://www.ine.es/en/

Author Response

  1. The effective reproduction number cannot be negative (l. 41). Probably the author means that it is below 1.

Reply: 

Thanks for pointing out this typo which has been corrected.

  1. It could be interesting to discuss the links between the CFR and the IFR. In particular, a lower bound for the IFR is the number of deaths divided by the total population in the region, and an upper bound is the CFR. 

Reply:

Thank you for this suggestion which has been incorporated into the penultimate paragraph of the article.

  1. The author indicates that he did not control for any confounders. It could be nice to check for a correlation between the median age (or the proportion of people older than some age) and the IFR in the considered regions. Such data can be found here : https://www.ine.es/en/

Reply:

In the new version the analyses are reported unadjusted and adjusted for the proportion of the population 65 years and older.

Reviewer 2 Report

Chris Kenyon presents an analysis on the serological and epidemiological data collected across different regions in Spain to provide estimations on the infection fatality rate (IFR) experienced by the different Spanish locations and the whole country. Despite the provided data is timely and the conclusions have great value to face the current health crisis, various concerns need to be addressed before this work should be considered for publication:

Major concerns:

1-"Epidemic intensity" is not defined, then it is difficult to understand the main hypothesis of the study. For instance, if "epidemic intensity" is defined as the number of cases per region or the number of deaths, then obviously more cases or more deaths influence the IFR because IFR is defined in terms of cases and deaths. In other words, the main hypothesis must be rewritten to assure the study is falsifiable.

1.1-Another issue with the compared variables is their independence. How independent are the variables shown in the figure? In other words, is there a way that these values can be independent? Is it possible to have high IFR and very low CFR, seroprevalence, cases, mortality or hospitalizations? In this study, these are correlated, but from the start that seems the expectation.

2-The origin of the data is not clear. If serological data was obtained, an ethics approval statement should be included or some explanation should be provided as why is not required. If the study is based on public data, the source and its accountability should be provided to assure the used data was rigorously compiled.

3-Among the limitations of the study, it is mentioned that demographic factors were not controlled for the estimations; however, the same discussion opens the door for some speculation. My suggestion is just to add the population distribution by age range (0-15, 15-60, >60) by autonomous community and provide that dimension to the study. Here is the link to the Spanish Statistical Office: https://www.ine.es/en/index.htm

Minor issues:

1-A minor but unsettling point is that there is only one author, but the text is written as "we". I suggest stating "I" or writing "this study...".

2-In the first paragraph, add the sample size of the cited study between parentheses: "One of the best quality estimates of IFR was determined in a relatively heavily affected community in Germany by accurately determining cause of death and establishing exposure via combined serological and nucleic acid amplification testing of a representative sample of the population (N=###)[3]."

3-The title states "ecological analysis", however, none of the ecological factors of these communities are discussed in the study.

Author Response

Major concerns:

1-"Epidemic intensity" is not defined, then it is difficult to understand the main hypothesis of the study. For instance, if "epidemic intensity" is defined as the number of cases per region or the number of deaths, then obviously more cases or more deaths influence the IFR because IFR is defined in terms of cases and deaths. In other words, the main hypothesis must be rewritten to assure the study is falsifiable.

Reply:

Thank you for pointing this ambiguity out. In the new version I have defined the 4 dimensions of epidemic intensity that are assessed (Page 3, Last sentence):

In this paper I hypothesized that IFR would similarly be influenced by 4 dimensions of epidemic intensity – seroprevalence of SARS-CoV-2, incidence of COVID-19 cases, COVID-19 mortality and COVID-19 hospitalizations.

1.1-Another issue with the compared variables is their independence. How independent are the variables shown in the figure? In other words, is there a way that these values can be independent? Is it possible to have high IFR and very low CFR, seroprevalence, cases, mortality or hospitalizations? In this study, these are correlated, but from the start that seems the expectation.

Reply:

This is an important question that I have not been able to delve into in this letter. There is however a fair amount of evidence that the relations between these variables varies between different populations and over time. In the introduction of the letter I note that the CFR in Wuhan declined considerably due to improved care [1]. IFRs and CFRs in countries with a younger populations are expected to be lower and preliminary data suggests this may be the case. There wide variations in case ascertainment and how deaths due to COVID-19 are classified. These are just some of the possible reasons why CFRs and IFRs may vary between populations. 

2-The origin of the data is not clear. If serological data was obtained, an ethics approval statement should be included or some explanation should be provided as why is not required. If the study is based on public data, the source and its accountability should be provided to assure the used data was rigorously compiled.

Reply:

The origin of the data is described in detail in the online supplement. This is stated in the paper as follows (Line 79):

See online supplement for complete list of sources of information as well as definitions of variables used.

New text has been added to address the fact that ethical clearance was not required (L 81-82).

All the data sources are open-access and only aggregated ecological level data was used. As such no ethical clearance was required.

3-Among the limitations of the study, it is mentioned that demographic factors were not controlled for the estimations; however, the same discussion opens the door for some speculation. My suggestion is just to add the population distribution by age range (0-15, 15-60, >60) by autonomous community and provide that dimension to the study. Here is the link to the Spanish Statistical Office: https://www.ine.es/en/index.htm

Reply:

As detailed in response to Reviewer 1, new analyses have been added controlling for age structure as advised by the reviwer.

Minor issues:

1-A minor but unsettling point is that there is only one author, but the text is written as "we". I suggest stating "I" or writing "this study...".

Reply:

The first person singular has been used where ever appropriate in the new version.

2-In the first paragraph, add the sample size of the cited study between parentheses: "One of the best quality estimates of IFR was determined in a relatively heavily affected community in Germany by accurately determining cause of death and establishing exposure via combined serological and nucleic acid amplification testing of a representative sample of the population (N=###)[3]."

Reply:

This has been added.

3-The title states "ecological analysis", however, none of the ecological factors of these communities are discussed in the study.

Reply:

It is true that there are a number of meanings for the term “ecological analysis”. In this analysis the term is used to a population level/ecological level analysis. Here is the formal definition of an ecological study in this context: 

"An ecological study is an observational study defined by the level at which data are analysed, namely at the population or group level, rather than individual level. Ecological studies are often used to measure prevalence and incidence of disease, particularly when disease is rare" [2].

References

  1. Zhang Z, Yao W, Wang Y, Long C, Fu X. Wuhan and Hubei COVID-19 mortality analysis reveals the critical role of timely supply of medical resources. J Infect. 2020. Epub 2020/03/27. doi: 10.1016/j.jinf.2020.03.018. PubMed PMID: 32209384.
  2. Levin KA. Study design VI - Ecological studies. Evid Based Dent. 2006;7(4):108. Epub 2006/12/26. doi: 10.1038/sj.ebd.6400454. PubMed PMID: 17187048.

Reviewer 3 Report

Using data provided by the Spanish ministry of health, the study by Kenyon determines the SARS-CoV-2 infection fatality rate (IFR) for the 19 Spanish autonomous communities. The author found a considerable variation in IFRs and that high IFRs are correlated with indicators for a rapid and extensive spread of SARS-CoV-2. The author concludes that the SARS-CoV-2 IFR is not fixed and that slowing the spread of COVID-19 reduces the IFR.

I think this short study should be considered for publication because it highlights the importance of strategies to slow down SARS-CoV-2 spread. However, the following questions/suggestions should be addressed before publication:

Headline: What does the author mean with ecological analysis?

Supplementary file: The hyperlinks provided in the reference section are incorrect because all dots are missing. Please correct them.

In my opinion the author should mention in the text that based on his results the idea of reaching SARS-CoV-2 herd immunity in a short time is a bad idea because it will result in a high IFR.

Author Response

I think this short study should be considered for publication because it highlights the importance of strategies to slow down SARS-CoV-2 spread. However, the following questions/suggestions should be addressed before publication:

Headline: What does the author mean with ecological analysis?

Reply:

In this analysis the term ecological analysis is used to a population level/ecological level analysis. Here is the formal definition of an ecological study in this context:

An ecological study is an observational study defined by the level at which data are analysed, namely at the population or group level, rather than individual level. Ecological studies are often used to measure prevalence and incidence of disease, particularly when disease is rare.

Reference:

Levin KA. Study design VI - Ecological studies. Evid Based Dent. 2006;7(4):108. Epub 2006/12/26. doi: 10.1038/sj.ebd.6400454. PubMed PMID: 17187048.

Supplementary file: The hyperlinks provided in the reference section are incorrect because all dots are missing. Please correct them.

Reply:

Thank you for pointing this out. The links have been corrected.

In my opinion the author should mention in the text that based on his results the idea of reaching SARS-CoV-2 herd immunity in a short time is a bad idea because it will result in a high IFR.

Reply:

This text has been added as suggested (L130-132).

Round 2

Reviewer 2 Report

In the revised version of the manuscript Chris Kenyon has addressed my previous concerns in a satisfactory revision. Following a couple of minor comments on the revised manuscript:

1-Consider removing "ecological" from the title, as the environment or interactions of the patients with their ecology plays no role in the analysis or discussion.

2-Please clarify in the following sentence that the results of the seroprevalence survey correspond to the same date: "The IFR per region was calculated as the percent of individuals who died from COVID-19 as of 13/5/2020 divided by the percent of individuals seropositive for COVID-19 in the seroprevalence survey."

3-In line 123, "The percent 65 or older ...", maybe it is better worded as "The percent of people with age 65 y. o. or older..."

4-In lines 127-133, I agree with the idea, but that sentence can be shortened and improve the message. A simple suggestion is: "Our results are consistent with the evidence that the incidence of COVID-19 is higher in elderly populations [15]. Since severe COVID-19 is characterized by high viral load in respiratory secretions, and this translates into increased infectiousness, vulnerable older populations confined to close spaces, such as elderly-care-type facilities, may have higher COVID-19 incidence [16]."

5-In figure 1, it could help to add the regression/trend lines to the panels to highlight the positive trend.

6-Multiple times the term "flattening the curve" is mentioned in the text, it would help the discussion to mention the main measures behind this policy such as social distancing, cocooning practices of elderly citizens, mask usage, etc.

Author Response

Reply to reviewer

1-Consider removing "ecological" from the title, as the environment or interactions of the patients with their ecology plays no role in the analysis or discussion.

Reply:

The word ‘ecological’ has been replaced with ‘population-level’

2-Please clarify in the following sentence that the results of the seroprevalence survey correspond to the same date: "The IFR per region was calculated as the percent of individuals who died from COVID-19 as of 13/5/2020 divided by the percent of individuals seropositive for COVID-19 in the seroprevalence survey."

Reply:

This sentence has been changed to the following:

The IFR per region was calculated as the percent of individuals who died from COVID-19 as of 13/5/2020 divided by the percent of individuals seropositive for COVID-19 in the seroprevalence survey (conducted 27/04/2020 and 11/05/2020).

3-In line 123, "The percent 65 or older ...", maybe it is better worded as "The percent of people with age 65 y. o. or older..."

Reply:

Thanks for this suggestion. This has been changed as suggested.

4-In lines 127-133, I agree with the idea, but that sentence can be shortened and improve the message. A simple suggestion is: "Our results are consistent with the evidence that the incidence of COVID-19 is higher in elderly populations [15]. Since severe COVID-19 is characterized by high viral load in respiratory secretions, and this translates into increased infectiousness, vulnerable older populations confined to close spaces, such as elderly-care-type facilities, may have higher COVID-19 incidence [16]."

Reply:

Thank you for this useful edit which has been used.

5-In figure 1, it could help to add the regression/trend lines to the panels to highlight the positive trend.

Reply:

6-Multiple times the term "flattening the curve" is mentioned in the text, it would help the discussion to mention the main measures behind this policy such as social distancing, cocooning practices of elderly citizens, mask usage, etc.

Reply:

This has been included in a sentence at L130:

These findings thus support the findings from other studies that flatten-the-curve strategies such as physical distancing, test-treat-isolate, cocooning high risk populations reduce not only total number of cases and deaths but also the probability of deaths per infection [7, 13].